# Shared decision-making for non-operative management versus operative management of hip fractures in selected frail older adults with a limited life expectancy: a protocol for a nationwide implementation study

Miliaan L Zeelenberg,[1] Paulieke C Oosterwijk,[2] Hanna C Willems,[3] Taco Gosens,[4] Dennis Den Hartog,[1] Pieter Joosse,[5] Sverre A I Loggers,[5] Thomas MP Nijdam,[6] Ruth E Pel-Littel,[7] Suzanne Polinder,[8] Henk Jan Schuijt ![ORCID],[9] Hugo H Wijnen,[10] Detlef Van der Velde,[11] Esther M M Van Lieshout ![ORCID],[1] Michael H J Verhofstad,[1] NOM-Implementation study group

MLZ and PCO contributed equally.

For numbered affiliations see end of article.

**Correspondence to**
Professor Esther M M Van Lieshout;
e.vanlieshout@erasmusmc.nl

## ABSTRACT

**Background and purpose** Recent research has highlighted non-operative management (NOM) as a viable alternative for frail older adults with hip fractures in the final phase of life. This study aims to guide Dutch physicians and hospitals nationwide in a standardised implementation of shared decision-making regarding surgery or NOM in selected frail older adults with a hip fracture.

**Methods and analysis** The patient population for implementation includes frail older adults aged ≥70 years with an acute proximal femoral fracture, nursing home care or a similar level of care elsewhere and at least one additional criterion (ie, malnutrition, severe mobility impairment or ASA≥4). The 2-year implementation study will be conducted in four phases. In phases 1 and 2, barriers and facilitators for implementation will be identified and an implementation protocol, educational materials and patient information will be developed. Phase 3 will involve an implementation pilot in 14 hospitals across the Netherlands. The protocol and educational material will be improved based on healthcare provider and patient experiences gathered through interviews. Phase 4 will focus on upscaling to nationwide implementation and the effect of the implementation on NOM rate will be measured using data from the Dutch Hip Fracture Audit.

**Ethics and dissemination** The study was exempted by the local Medical Research Ethics Committee (MEC-2023-0270, 10 May 2023) and Medical Ethics Committee United (W23.083, 26 April 2023). The study's results will be submitted to an open access international peer-reviewed journal. Its protocols, tools and results will be presented at several national and international academic conferences of relevant orthogeriatric (scientific) associations.

**Trial registration number** NCT06079905.

## STRENGTHS AND LIMITATIONS OF THIS STUDY

⇒ This study builds on an existing foundation of recent well-received (inter)national publications, countrywide initiatives involving shared-decision making and recent demands for assistance in the implementation of non-operative management of hip fractures in selected frail older adults with a limited life expectancy.

⇒ The study will employ a staged approach in four phases, involving implementation strategies at the individual, local and national level, that range from local educational sessions to nationwide distribution at conferences and proposed guideline changes.

⇒ The study design allows for regular adjustment of its implementation protocols and educational materials based on patient and healthcare provider feedback from interviews from 14 different hospitals throughout the Netherlands.

⇒ While the study has an ambitious 2-year time scheme to achieve nationwide implementation in the hospital setting, it will provide an important foundation for further initiatives.

## INTRODUCTION

A hip fracture in frail older adults can indicate an approaching end of life. Advanced age, multiple comorbidities and severe mobility limitations are important risk factors for early postoperative mortality after a hip fracture.[1 2] Studies show that >30% of patients die within 1-year post hip fracture, with 6-month mortality rates in older adults with dementia exceeding 55%, often with low quality of life in the months preceding death.[3–5] These patients, in their final phase of life, often

do not benefit from an operative treatment that aims to restore mobility. Operations are often conducted in this population as a palliative treatment option, focused primarily on pain reduction. Based on these insights, a paradigm shift is occurring in clinical approach, emphasising the importance of individualised care plans focused on optimal quality of life.[6 7]

The FRAIL-HIP study has confirmed that non-operative management (NOM) can be a good alternative to operative management in select frail older adults with a limited life expectancy.[5 8–10] Quality of life after NOM was valued non-inferiorly and the majority of the next-of-kin rated the quality of dying of their relative as very good.[5 8] Despite higher mortality rates in patients opting for NOM, the option to allow patients to pass away in a familiar environment was a compelling factor in favour of NOM in this select population.[11] Both operative and NOM can thus be effective treatment options as part of palliative care in selected frail older adults with a limited life expectancy. Following these results, multiple Dutch hospitals have asked for guidance on implementing NOM in their institutions.

Central to NOM is the shared decision-making (SDM) process, involving patients and their families in treatment decisions aligned with the patients' preferences and needs. Based on the results of the FRAIL-HIP study and other literature, an implementation protocol has been drafted, piloted and refined based on experiences of healthcare professionals.[7 9 12] This protocol version will serve as the starting point for the iterative process towards nationwide implementation.

The aim of this study is nationwide implementation of SDM on treatment of hip fractures, involving both NOM and operative treatment, to ensure that frail older adults receive optimal, patient-centred care in the final phase of their lives.

## METHODS AND ANALYSIS
### Study design and aims
This 2-year implementation study will consist of four phases (figure 1): development of the implementation protocol and educational material (phase 1 and 2), implementation across a subset of hospitals across the Netherlands and process evaluation and optimisation of the protocol based on healthcare provider/patient experiences (phase 3) and upscaling towards nationwide implementation and project evaluation (phase 4). The study design of phases 3 and 4 entails a multicentre prospective cohort study. Overall, the project will cover all activities as recommended in the ImpRes Guide, which mentions four implementation phases (ie, exploration, installation, initial implementation and full implementation) and should efficiently translate local knowledge into general knowledge.[13] The 'Consolidated Framework for Implementation Research' (CFIR) was used for designing this project.[14 15] With education being a critical part of the implementation, elements of the education strategy from Wensing *et al*[16] were added. The CFIR domains that the four phases of this project will cover are mentioned in table 1.

To achieve the primary aim, this study will use a staged approach in four phases that involves three levels: he healthcare provider specific/local hospital level (micro) in phases 1–3, the level of a limited number of hospitals (meso) in phase 3 and the national level (macro) in phase

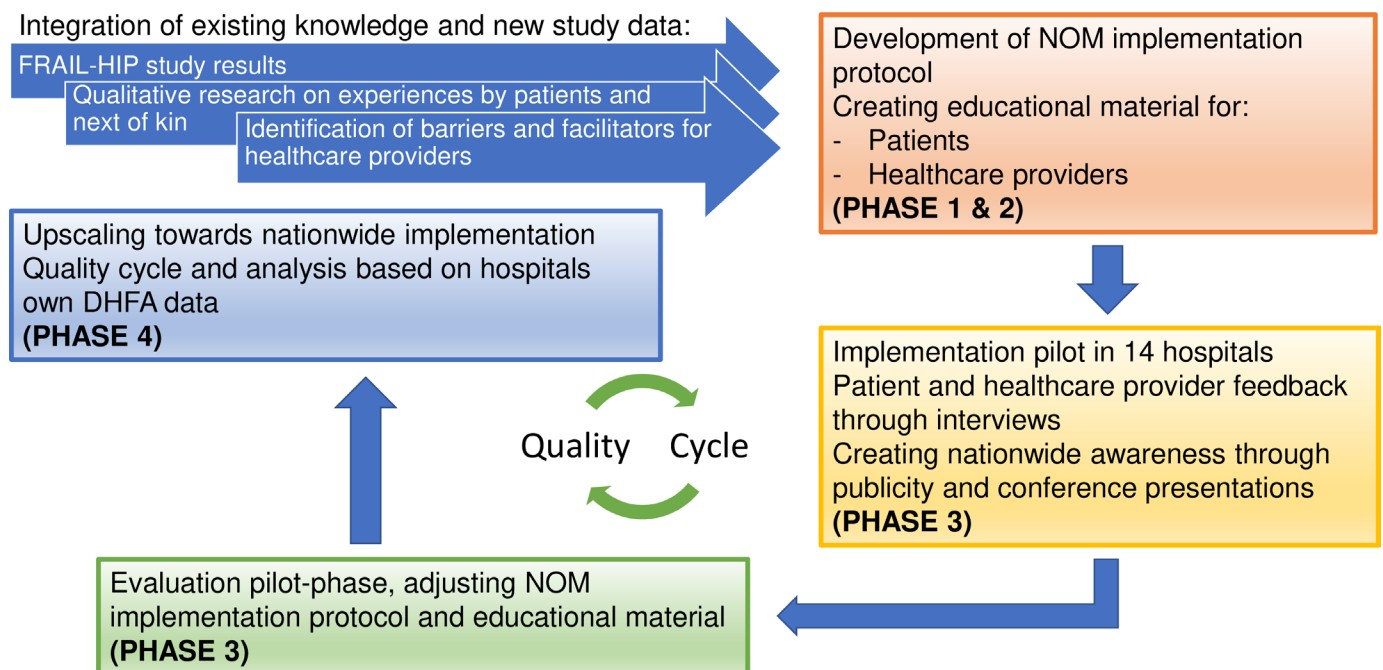

**Figure 1**  Schematic overview of the implementation process in four phases. DHFA, Dutch Hip Fracture Audit; NOM, non-operative management. FRAIL-HIP study[8]

**Table 1** CFIR domains and their role in the implementation project

| CFIR domain | Role in project |
|---|---|
| Inner setting | Gain insight into the hospital/implementation location, including internal network and communication, culture (norms and values), knowledge, current use of the intervention and implementation climate (tension for change, feedback, learning climate) |
| Outer setting | Gain insight into the needs of patients/proxies and pressure from peers (need for hospitals to use the same protocol) |
| Characteristics of individuals | Gain insight into the confidence/preparedness to discuss and apply NOM through SDM with the intended target group |
| Characteristics of the intervention | Gain insight into the SDM conversation as part of NOM, the needs of healthcare providers on how to conduct SDM and who participates in it, and what they need to apply NOM |

CFIR, Consolidated Framework for Implementation Research; NOM, non-operative management; SDM, shared decision-making.

4. The four consecutive phases of the project have phase-specific aims with corresponding target populations and points of action, that are described in table 2. To ensure nationwide implementation, many parties and interest groups are or will be involved in the project (table 3).

### Study population

Phases 1 and 2 will not include patients or patient data but will include healthcare provider and patient association perspectives. Phases 3 and 4 will also include frail older adults with a limited life expectancy who sustained a proximal femoral fracture. Patients will be included

and receive SDM on NOM and operation if they meet the following criteria:

1. Acute proximal femoral fracture (AO type 31), confirmed on X-ray or CT scan.
2. 70 years or older.
3. Living in a nursing home or receiving similar care at home.
4. And at least one of the following characteristics:
   – Malnutrition (cachexia or body mass index<18.5 kg/m$^2$).
   – Functional ambulation category[*] (FAC) 2 or lower.

**Table 2** Study aims with specific action points per targeted group for the four study phases

| Points of action per phase | Targeted group | Aim/outcome |
|---|---|---|
| **Phase 1** | | |
| Qualitative research on patient experiences and perceived barriers and facilitators for SDM on NOM | Patients, next-of-kin and healthcare providers* | Enriching the implementation protocol with recent research on patient/healthcare provider experiences regarding SDM on NOM versus operative treatment. Providing insight into barriers and facilitators on the local and individual level |
| **Phase 2** | | |
| Development of educational material on SDM with frail older adults with hip fractures for patients and healthcare providers Implementing SDM on NOM in the curriculum for (orthopaedic) surgeons in training | Patients, next-of-kin and healthcare providers* | Increasing (practical) knowledge on performing SDM on NOM in healthcare providers. Providing healthcare providers with tools to prepare patients and next-of-kin for SDM on NOM vs operative treatment |
| **Phase 3** | | |
| Presentations of the implementation protocol and included tools at annual meetings of the involved scientific associations | Healthcare providers*, members of associated scientific associations | Creating awareness and a sense of urgency for SDM implementation and identifying local drivers of change for future implementation |
| Implementation of the NOM implementation protocol and evaluation through interviews with patients and healthcare providers in 14 selected Dutch hospitals | Local project teams and drivers of change in 14 Dutch hospitals and patients/next-of-kin | Evaluating and improving the implementation protocol and educational modules based on local experiences, barriers and facilitators through patient and healthcare provider interviews |
| **Phase 4** | | |
| Maintaining and improving awareness through conference and regional trauma network presentations Identification of local ambassadors/drivers of change Dissemination of the implementation protocol/educational tools through a dedicated website, scientific publications and media coverage | Healthcare providers*, and specifically, local drivers of change, in all Dutch hospitals treating frail older adults with hip fractures | Upscaling of the implementation to all hospitals in the Netherlands that treat the targeted patient group |
| Measuring the effect of implementation, changes in NOM rate, through the DHFA database | Dutch patients and hospitals through the DHFA database | Evaluating changes in practice after implementation |

*Healthcare providers include (orthopaedic) trauma surgeons, geriatricians, acute care physicians, anaesthesiologists, residents and physician assistants directly involved in (acute) care for frail older adults with a hip fracture in Dutch hospitals.
DHFA, Dutch Hip Fracture Audit; NOM, non-operative management; SDM, shared decision-making.

**Table 3** Participating parties and their roles and responsibilities

| Participator(s) | Roles and responsibilities |
| --- | --- |
| Central project group/steering committee | The project group is responsible for the general oversight of the entire project. |
| Participating hospitals | They will implement the NOM protocol locally, and will provide data needed to evaluate the effect of the implementation. They will need to allow local training and participate in interviews to gain insight into their experience with NOM implementation. |
| Patients and/or proxies | Will be asked to voluntarily participate in interviews and provide feedback on the chosen treatment and SDM process. |
| Dutch Hip Fracture Audit (DHFA) and Dutch Institute for Clinical Auditing | They will provide the data so the hospitals can use it as mirror information. The DHFA data (which are existing data) will be used as preimplementation and postimplementation data, and will thus allow accurate monitoring of the impact of NOM implementation. |
| Scientific associations (Dutch Trauma Society (NVT), Dutch Association of Surgeons of the Netherlands (NVvH), Dutch Orthopaedic Association (NOV) and Dutch Association of Clinical Geriatrics (NVKG)) | The scientific associations are responsible for education for medical specialists and residents-in-training. They will play a key role in the education sessions and in forwarding information on our project to their members. |
| Members of the National Association of General Practitioners (LHV) and the Association for Older Adult Care Physicians (Verenso) | Will be kept informed about this project through mailings and presentations during their annual conferences. |

LHV, Landelijke Huisartsen Vereniging; NOM, non-operative management; NOV, Nederlandse Vereniging voor Orthopedie; NVKG, Nederlandse Vereniging voor Klinische Geriatrie; NVT, Nederlandse Vereniging voor Traumachirurgie; NVvH, Nederlandse Vereniging voor Heelkunde; SDM, shared decision-making.

– Severe comorbidities (American Society of Anesthesiologists (ASA) category 4 or 5).

Patients and/or proxies who do not have sufficient comprehension of the Dutch language will be excluded from the interviews during phase 3.

*The FAC is a functional walking test that evaluates ambulation ability.[17] This 6-point scale assesses ambulation status by determining how much human support the patient requires when walking, regardless of whether or not they use a personal assistive device. FAC≤2 means that a patients at least has the need for (intermittent) help of another person to be able to ambulate (FAC 2) to no functional ambulatory capabilities (FAC 0).

### Implementation in four phases
#### Phase 1 (exploration): evaluate current practice and identify facilitators and barriers of NOM implementation
Based on the evidence-based 'Measurement Instrument for Determinants of Innovations')[18 19] and the 'Barriers and Facilitators Assessment Instrument',[20] a survey has been developed, aimed to identify the facilitators and barriers for applying SDM and NOM for selecting the best treatment for individual patients of the target population and identified five barriers and 23 facilitators.[9] The patient perspective was not included in this study but was explored in another study.[11] A panel from older adults organisations, general practitioners and geriatric specialists will discuss and supplement the data from the aforementioned studies from their perspective.

#### Phase 2 (installation): developing an implementation strategy for NOM
A draft protocol for NOM implementation, with as focal point the SDM interview, has been written based on the experience and expertise of the project members, previous studies and questionnaires. This protocol contains the central vision, the requirements for the implementation of NOM in selected frail older adults and a recommendation for the composition of a local project team of stakeholders and drivers of change. To remove barriers, practical tools such as pocket cards, posters, factsheets and local training options will be drafted or updated by the project team, based on the 'Expert Recommendations for Implementing Change strategies'.[21] This should help drivers of change in achieving implementation goals and change day-to-day policies. The NOM implementation protocol will contain an overview of the data to be collected for NOM evaluation. A reporting tool will be developed in the Dutch Hip Fracture Audit (DHFA) for this purpose. In addition, educational modules for healthcare professionals will be developed using the NOM protocol in combination with existing educational modules on SDM.[22–24] NOM as a treatment option for selected frail older adults will also be included in the obligatory curriculum for residents by the Association of Surgeons of the Netherlands (in Dutch: Nederlandse Vereniging voor Heelkunde), in order to incorporate NOM into the educational programme of (orthopaedic) trauma residents.

#### Phase 3 (initial implementation): NOM-implementation in selected hospitals
NOM for selected frail older adults will be implemented in 14 selected hospitals (figure 2, collaborators listed in online supplemental materials 1). Based on the experience of these hospitals, the NOM implementation protocol and/or the educational material will be optimised. To safeguard implementation of NOM in the long term, the NOM protocol pays explicit attention to the involvement of all relevant healthcare providers by informing those involved before, during and after implementation, and by collecting data on relevant feasibility parameters as described in the Bowen model.[25] To achieve this, the site principal investigator will be asked to establish a local project team. The team should represent all groups involved in NOM, including relevant physicians and, as applicable, residents of the involved departments,

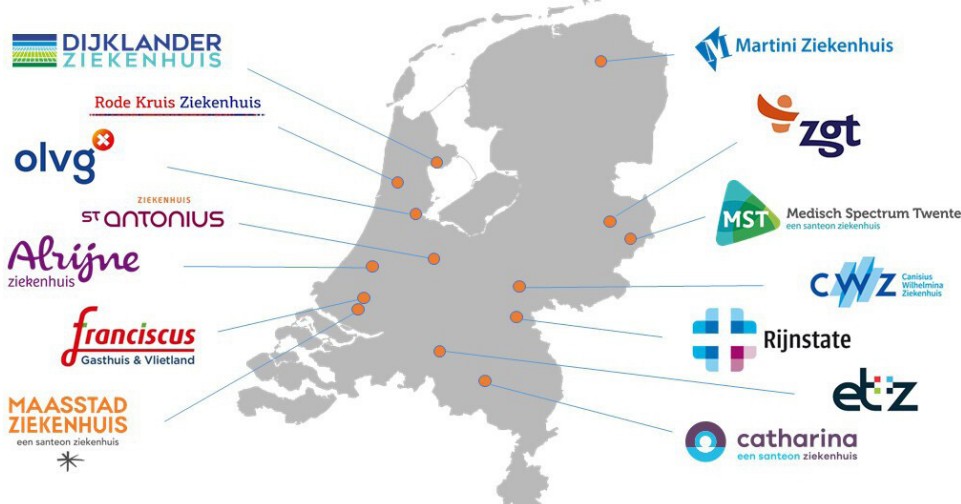

**Figure 2** Map of participating hospitals during the initial implementation in selected hospitals (phase 3).

specialised nurses, physician assistants and hospital management. Prior to implementation, this local team will meet with the researchers for an interview to train them in applying the SDM and the NOM implementation protocol, and to collect details on their specific needs for the implementation.

The local team will perform patient selection and apply SDM on NOM and operative treatment for all patients who meet the eligibility criteria. After the start of the implementation, during the first 6 months, patients, relatives and/or next-of-kin will be approached once by telephone to ask about their satisfaction with the treatment decision process and the chosen treatment. This will be done through a semistructured interview. To provide more insight about their own patient population, the interview results will be discussed with the participating centres on a quarterly basis. All collected data will be used to improve the NOM protocol and/or educational material prior to phase 4. Six months after start of implementation, the local team will be approached again for an interview to evaluate the process with the relevant parameters: adoption, integration, applicability, expansion and early local effectiveness.

### Phase 4 (nationwide implementation): NOM-implementation in all hospitals in the Netherlands and project analysis

Phase 4 is an expansion of phase 3 to reach nationwide implementation. The implementation strategy will essentially be the same as in phase 3. Due to the increase in number of hospitals, the supervision and contact with the researchers and steering committee will be less intensive than in phase 3. In practice, this means that the NOM protocol will be implemented, but the interviews with healthcare professionals in the hospitals (prior to start and 6 months after start) will be omitted. To ensure that all information on SDM and NOM implementation is available continuously to both patients and healthcare providers, a dedicated website will be developed and maintained. To effectively reach a broad audience of

healthcare providers and keep them updated on the implementation process and learning experiences, our goal is to give presentations at the annual meetings of orthogeriatric scientific associations, as well as associations for general practitioners and elderly care physicians (table 3). In addition, time slots for presentations will be requested at the meetings of the regional trauma networks. Ultimately, we aim to include SDM on NOM or operative treatment for proximal femoral fractures in select frail older adults in the National guidelines for hip fracture treatment and operative care in older adults.

### Outcome measures and data collection

The primary outcome measure (for phases 3 and 4) is (change in) the NOM rate. Type of treatment (NOM vs operative treatment) will be extracted from the DHFA. The DHFA is a national database, including over 95% of Hip fracture patients and can provide automated anonymous patient data up to 1 year after hip fracture treatment. To evaluate the effect of the implementation, the change in NOM rate and operative rate for the targeted population will be evaluated and compared with the rate prior to implementation. This will be calculated using the annual national DHFA data from the 2 years preceding implementation per hospital, as baseline. The postimplementation NOM rate will be collected immediately after completion of phase 4, as well as at the end of 2025. The secondary outcome measures (for phase 3 only) will be collected during semistructured interviews with patients/proxy or next-of-kin, 4–6 weeks after inclusion. These are the satisfaction with the chosen treatment on a numeric rating scale (0–10), their valuation of the information received during the SDM process, and if the patient's needs were respected enough in the treatment decision. The interview sheets (in Dutch), used for patient or healthcare provider interviews in phases 1–3, are included in online supplemental materials 2.

The following patient data will be also collected in phases 3 and 4 from the DHFA at 3 months and 1 year

after hip fracture treatment (emerging from the patients' medical records). Patient characteristics (age, sex, ASA, FAC), prefracture situation (residence, mobility, Katz Activities of Daily Living (ADL) score), fracture details (fracture classification, affected side), hospital details (hospital length of stay and intensive care length of stay) and outcome details (complications, revision surgeries, readmission, 30-day mortality and time until death).

Data will be collected for each patient who meets the eligibility criteria. To reduce the administrative burden, the DHFA database will be used as much as possible. Data from the DHFA will be imported in a Castor EDC database, in which hospitals can only see their own data. The datasets of all hospitals will be combined for the final analysis.

## Sample size calculation

A formal sample size calculation was not made as the primary aim is implementation on a national level. To give an idea of the total national population size the following estimate was made: In 2020, 21 165 patients presented to a hospital in the Netherlands with a proximal femoral fracture.[26] Of these 17 225 (81.5%) were aged 70 years or older. An approximate 25% of this population lived in a nursing home or other institution. Which projects to about 4300 patients as the primary target population of this study. Approximately half of these patients will meet our inclusion criteria and thus not all patients will be eligible for NOM, but this gives a broad estimate of the annual number of patients that should at least be considered for SDM on NOM and operative treatment.

## Statistical analysis

Data will be analysed by using the SPSS V.28 or higher (SPSS). Normality of continuous data will be tested with the Shapiro-Wilk test, and homogeneity of variances will be tested using the Levene's test. A $p<0.05$ will be taken as threshold of statistical significance. Missing values will not be imputed.

The primary outcome, NOM rate, will be determined by dividing the number of patients treated with NOM by the total eligible number of patients (per participating centre and for all participating centres combined).

All continuous data will be reported as mean with standard deviation (SD; if normally distributed based on Shapiro-Wilk test) or as median with quartiles (if not normally distributed). Categorical data will be reported as number with percentage. Preimplementation and postimplementation data will be compared with a Student's t-test (continuous data, normally distributed), Mann-Whitney U-test (continuous data, non-normally distributed), or a $\chi^2$ or Fisher's exact test (categorical data), as applicable. The analysis will be stratified by NOM versus operative treatment, and further stratified as feasible depending on the final study population, for example, by time since NOM implementation, prefracture mobility, ADL status, fracture classification or bilateral fracture.

## Project timeline

Phase 1 of the study started in June 2023 and ended in August 2023. Phase 2 started in June 2023 and ended in October 2023. Phase 3 started in September 2023 and will approximately end in October 2024. Phase 4 will start in June 2024 and will approximately be concluded in June 2025.

## Patient and public involvement

A patient association representative was consulted during the drafting of the study protocol and could provide input for the study design and implementation tools. Through interviews, study participants (patients, next-of-kin and healthcare providers) will be directly involved in providing feedback on the SDM process and educational tools used during the project. In addition to general publication, a summary of the main results will be made available to study participants on request.

## Ethics and dissemination

### Ethical approval, risks and dissemination

The study was exempted by the local Medical Research Ethics Committee (MEC-2023–0270, 10 May 2023) and Medical Ethics Committee United (W23.083, 26 April 2023) and is registered at ClinicalTrial.gov (NCT06079905, 11-10-2023). The study was approved by the board of directors of all hospitals participating in phase 3. There are no risks involved in this study. The only burden is the time needed for the interviews with healthcare providers and patients/next-of-kin in phase 3. The interviews will be performed by researchers following an opt-out design, participants can stop at any time. Participants will be asked to consent to share their contact details with the researchers and participation means interview data can be used (online supplemental materials 3). In addition, only already existing and routinely collected data will be collected. All treatments, chosen after SDM, will be performed following local standard care protocols. Based on the above, the overall burden and risks are considered negligible. The study will follow Good Clinical Practice guidelines. The study's results will be submitted to an open access international peer-reviewed journal. As previously explained above, its protocols, tools and results will be made publicly available and presented at several national and international academic conferences of relevant orthogeriatric (scientific) associations.

### Data management, monitoring and privacy

All data and information collected regarding this study will be treated confidentially by the researchers. Research data that can be traced to individual persons can only be viewed by authorised personnel (members of the research team and members of the healthcare inspection). Data will be encoded and stored in a Castor EDC database with password-restricted access to the researchers at individual sites only. Regarding participants privacy, the Dutch 'General data protection regulation implementation act' will be applied.

## DISCUSSION
### Clinical impact

This project provides an important translation of recent scientific publications to clinical practice and its success could play an important role in (inter)national surgical and geriatric guideline changes. Successful implementation of NOM will change the way healthcare providers view acute hip fractures in frail older adults with a limited life expectancy. Instead of primarily focusing on clinical possibility of hip fracture treatment, the focus will shift towards treatment concerning patients' quality of life, harm reduction and effective palliative care in the final stages of life. Because of this shift, the NOM rate is expected to rise in this select group. However, the main goal is not to increase NOM rate but to provide each patient with the clinical care that best suits their wishes, needs and clinical reality. Nationwide implementation of SDM on NOM will also lead to a reduction of overtreatment and might reduce associated costs.[10] Another important outcome is reduced practice variation between hospitals in both patient selection for (SDM on) NOM and nationwide implementation of a universal structured SDM protocol for frail older adults with a hip fracture.

### Strengths and limitations

The strength of this study is that it builds on an existing foundation of recent well-received (inter)national publications, countrywide initiatives involving SDM and advance care planning, and recent demands for assistance in the implementation of NOM from several hospitals. The project group includes experienced professionals from (orthopaedic) trauma surgery, geriatrics, epidemiology and implementation science from both university medical centres and general hospitals, supplemented with patient perspectives and experts in SDM and study methodology. A limitation of the project is the limited time period for the initial project. Achieving nationwide implementation within 2 years, which is the maximum allowed project duration of the grantor, might be an overoptimistic aim. However, the project will have developed a strong foundation on which future projects can build and will keep sharing its tools and information, even after completion. Another limitation is that the inclusion criteria might prove to be too narrow. There might be a larger group of patients for whom NOM could suit their preferences or goals of care. However, the project group has decided to keep strict criteria to prevent the pendulum swinging too far and causing undertreatment with long-term survival without functional recovery after opting for non-operative treatment, which results in potential decreased quality of life and associated adverse events. Depending on the results of the initial implementation, the inclusion criteria in the protocol might be adjusted to suit the needs of patients and healthcare providers. Finally, the NOM rate in the DHFA database might be an underestimation, as patients living in a nursing home might already have advance care plans preventing them from being transferred to the hospital in the case of a hip fracture. As the NOM implementation gains national traction more attention to advance care planning around hip fractures might lead to an unmeasured increase of the NOM rate in the prehospital setting. While reducing DHFA registration for this project, improved communication on patients wishes and integration of hip fractures in advance care planning, between patients and (geriatric) medical practitioners within and outside the hospital, is a desired effect that will improve the quality of NOM and patients' treatment satisfaction.

### Future perspective

The success of this project can be used as an international example for trauma care in selected frail older adults and lead to further changes in advance care planning within this older trauma population. The projects finished protocols and clinical tools may be used for updating the Dutch national 'Guideline for the treatment of proximal femoral fractures' and 'Guideline for multidisciplinary treatment of operative treatment of frail older adults' and may also help updating international treatment guidelines on proximal femoral fractures in selected frail older adults with a limited life expectancy.

**Author affiliations**
[1]Trauma Research Unit Department of Surgery, Erasmus MC, University Medical Center Rotterdam, Rotterdam, the Netherlands
[2]Department of Trauma Surgery, Sint Antonius Ziekenhuis, Utrecht, the Netherlands
[3]Department of Geriatrics, Amsterdam University Medical Center, Amsterdam, the Netherlands
[4]Department of Orthopaedic Surgery, Elisabeth-TweeSteden Ziekenhuis, Tilburg, the Netherlands
[5]Department of Trauma Surgery, Noordwest Ziekenhuisgroep, Alkmaar, the Netherlands
[6]Department of Surgery, Sint Antonius Ziekenhuis, Nieuwegein, the Netherlands
[7]Vilans, Utrecht, the Netherlands
[8]Department of Public Health, Erasmus MC, University Medical Center Rotterdam, Rotterdam, the Netherlands
[9]Department of Trauma Surgery, Sint Antonius Hospital, Utrecht, the Netherlands
[10]Department of Clinical Geriatrics, Rijnstate, Arnhem, the Netherlands
[11]Department of Surgery, Sint Antonius Ziekenhuis, Utrecht, the Netherlands

**Collaborators** The 'NOM-Implementation study group' is also included in online supplementary material 1. These include: E. Sohl, J.A. Jansen, M. Leijnen, W.A.H. Van der Stappen, J. Jansen, M. Peters-Kop, A.H. Van der Veen, N.C. Schepel, J.A.M. Wilmer, J. Steens, J. Winkelhagen, N.M.F. Noorda, H.A.A.M. Maas, O. Wijers, V. Vis, D. Van der Stap, G.R. Roukema, E. Bosma, T.M. Van Raaij, A.M. Van der Knaap, R. De Groot, A.V.C.M. Zeegers, H.A. Formijne Jonkers, D.H.R. Kempen, K. De Vries, A.F. Pull ter Gunne, M.P. Somford, E. Tanis, J.H. Duits, R.A. Zandbergen, N.C. Leegwater, O.C. Geraghty, J.H. Hegeman, E.M. Regtuijt.

**Contributors** MLZ, PCO, HW, TG, DDH, PJ, SAIL, TMPN, REP-L, SP, HJS, HHW, DVdV, EMMVL and MHJV were all involved in conceptualisation and/or designing of the study, and the writing and revising of associated documents and protocols. MHJV will act as principal investigator. MLZ and PCO wrote the draft of the main manuscript. EMMVL and HJS provided direct supervision. MLZ, PCO, HW, TG, DDH, PJ, SAIL, TMPN, REP-L, SP, HJS, HHW, DVdV, EMMVL and MHJV actively contributed to critical revision of the manuscript in multiple revision rounds. The NOM-Implementation study group contains the representatives from each of the 14 participating hospitals. All authors have read and approved the final version of this manuscript.

**Funding** This project was previously peer-reviewed and accepted for funding by the Netherlands Organisation for Health Research and Development (ZonMw; project No. 10390162210008).

**Competing interests** None declared.

**Patient and public involvement** Patients and/or the public were involved in the design, or conduct, or reporting, or dissemination plans of this research. Refer to the Methods section for further details.

**Patient consent for publication** Not applicable.

**Provenance and peer review** Not commissioned; externally peer reviewed.

**ORCID iDs**
Henk Jan Schuijt http://orcid.org/0000-0002-7399-6828
Esther M M Van Lieshout http://orcid.org/0000-0002-2597-7948

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
