## [Reviewer comments · BMJ Open]

ARTICLE DETAILS

TITLE (PROVISIONAL)	Shared decision making for nonoperative management versus operative management of hip fractures in selected frail older adults with a limited life expectancy; a protocol for a nationwide implementation study
AUTHORS	Zeelenberg, Miliaan; Oosterwijk, Paulieke; Willems, Hanna; Gosens, Taco; Den Hartog, D; Joosse, Pieter; Loggers, Sverre; Nijdam, Thomas; Pel-Littel, Ruth; Polinder, Suzanne; Jan Schuijt, Henk; Wijnen, Hugo; Van der Velde, Detlef; Van Lieshout, Esther M.M.; Verhofstad, Michael; , PNOM-Implementation Study Group

VERSION 1 – REVIEW

REVIEWER	R. Buckley Foothills Medical Centre
REVIEW RETURNED	10-Jan-2024

GENERAL COMMENTS	Good paper - no revisions requested.
--------------------------------------

REVIEWER	Paul Linsley University of East Anglia, Health Sciences
REVIEW RETURNED	24-Jan-2024

GENERAL COMMENTS	From the protocol it is immediately evident what the trial is investigating. It provides a detailed overview of the proposed study in line with its county's and organisation guidelines for protecting the safety of its subjects and meets ethical considerations and requirements. The methodology and methods are robust and in line with the the study aim and objectives. The rationale and background information provide specific reasons for conducting the research in light of pertinent knowledge about the research topic. The study design justifies the scientific integrity and credibility of the research study. This is a clear and well written protocol which other researchers could run with. I wish you success in your study.
--

REVIEWER	Mriganka Singh Brown University
REVIEW RETURNED	29-Jan-2024

GENERAL COMMENTS	A thoughtfully designed protocol for a nationwide implementation of shared decision making intervention in frail older adults with proximal hip fractures and limited life expectancy. Methods section: This is a planned study protocol. BMJ guidelines require inclusion of planned study dates in the protocol. Adding the dates for each of the 4 phases would enhance the study protocol. Outcomes: The study protocol appears to be using a pre-test post test design. A few aspects need more clarification
--

	1. What is the timeframe and the dates when the baseline or pre-intervention PNOM (primary outcome) will be collected? 2. Will the baseline PNOM rate be collected once? More than one observation may make the design stronger. 3. How many months after implementation will the post intervention PNOM rate be collected? Will it be collected once or more than once? 4. Is there a possibility of collecting pre-intervention and post-intervention Operative rates as well in addition to PNOM rates? 5. For outcome details, what is the observation window for complications, revision surgeries and readmissions?
--	--

VERSION 1 – AUTHOR RESPONSE

Reviewer Questions/Requested changes	Author Response/Manuscript adjustments
Reviewer 1:	
Good paper - no revisions requested.	-
Reviewer 2:	
This is a clear and well written protocol which other researchers could run with. I wish you success in your study.	-
Reviewer 3:	
This is a planned study protocol. BMJ guidelines require inclusion of planned study dates in the protocol. Adding the dates for each of the 4 phases would enhance the study protocol.	A paragraph titled "Project timeline" was added to the manuscript method section. It includes the approximate dates for the four phases of the study (Page 15).
What is the timeframe and the dates when the baseline or pre-intervention PNOM (primary outcome) will be collected?	The pre-intervention PNOM and operative rates will be collected based on the DHFA data from the two years preceding implementation: 2022 and 2021. This information was added to the manuscript on page 13, line 10-13.
Will the baseline PNOM rate be collected once? More than one observation may make the design stronger.	The baseline PNOM rate will be collected for the two years preceding implementation, as mentioned in the previous response. This information was added to the manuscript on page 13, line 10-13.
How many months after implementation will the post intervention PNOM rate be collected? Will it be collected once or more than once?	The PNOM rate will be collected at the end of the project in May/June 2025. To ensure complete DHFA data the data will be collected again on completion of the DHFA for 2025. This information was added to the manuscript on page 13, line 12-14.

Is there a possibility of collecting pre-intervention and post-intervention Operative rates as well in addition to PNOM rates?	Operative rates will also be collected from the DHFA data. This was described more clearly on manuscript page 13, line 9.
For outcome details, what is the observation window for complications, revision surgeries and readmissions?	The observation window for outcome details will be a maximum of 1 year after hip fracture treatment. As mortality is very high in this population most outcomes are expected within the first 3 months. This information was added to the manuscript on page 13, line 7-8 and 20-21.

VERSION 2 – REVIEW

REVIEWER	Mriganka Singh Brown University
REVIEW RETURNED	05-Mar-2024
GENERAL COMMENTS	Revisions noted. The change from PNOM to NOM is better suited as well.